# Electroporation Induces Unexpected Alterations in Gene Expression: A Tip for Selection of Optimal Transfection Method

**DOI:** 10.3390/cimb47020091

**Published:** 2025-01-31

**Authors:** Taiji Hamada, Seiya Yokoyama, Toshiaki Akahane, Kei Matsuo, Ikumi Kitazono, Tatsuhiko Furukawa, Akihide Tanimoto

**Affiliations:** 1Department of Pathology, Graduate School of Medical and Dental Sciences, Kagoshima University, 8-35-1 Sakuragaoka, Kagoshima 890-8544, Japan; 2Center for Human Genome and Gene Analysis, Kagoshima University Hospital, 8-35-1 Sakuragaoka, Kagoshima 890-8544, Japan; 3Department of Surgical Pathology, Kagoshima University Hospital, 8-35-1 Sakuragaoka, Kagoshima 890-8544, Japan; 4Center for Research of Advanced Diagnosis and Therapy of Cancer, Graduate School of Medical and Dental Sciences, Kagoshima University, 8-35-1 Sakuragaoka, Kagoshima 890-8544, Japan

**Keywords:** genome editing, electroporation, *PDGFRA*, receptor tyrosine kinase genes, lipofection, recombinant adeno-associated virus

## Abstract

Electroporation is an efficient method for nucleotide and protein transfer, and is used for clustered regularly interspaced short palindromic repeat (CRISPR)-associated protein 9 (Cas9)-mediated genome editing. In this study, we investigated the effects of electroporation on platelet-derived growth factor receptor alpha (*PDGFRA*) and receptor tyrosine kinase (*RTK*) expression in U-251 and U-87 MG cells. *PDGFRA* mRNA and protein expression decreased 2 days after electroporation in both cell lines, with recovery observed after 13 days in U-87 MG cells. However, in U-251 MG cells, PDGFRα expression remained suppressed, despite mRNA recovery after 13 days. Similar expression profiles were observed for lipofection in the U-251 MG cells. Comprehensive RNA sequencing confirmed electroporation-induced up- and down-regulation of *RTK* mRNA in U-251 MG cells 2 days post-electroporation. In contrast, recombinant adeno-associated virus (rAAV) transfected with mNeonGreen fluorescent protein or Cas9 did not affect *PDGFRA*, *RTKs*, or inflammatory cytokine expression, suggesting fewer adverse effects of rAAV on U-251 MG cells. These findings emphasize the need for adequate recovery periods following electroporation or the adoption of alternative methods, such as rAAV transfection, to ensure the accurate assessment of CRISPR-mediated gene editing outcomes.

## 1. Introduction

Genome editing using the clustered regularly interspaced short palindromic repeat (CRISPR)-associated protein 9 (Cas9) is a widely used tool for genetic modification in many organisms [1,2,3]. Gene knockouts and knock-ins using the CRISPR/Cas9 system in cultured cells enable functional studies of genes and proteins, providing valuable insights into gene function [4]. To edit a gene of interest, Cas9 is introduced into cells through an expression plasmid, mRNA, or protein [4]. Various transfection methods have been employed to introduce Cas9 or other genetic material into cells, including electroporation, which is one of the most efficient methods for delivering nucleic acids and proteins [5,6]. Electrical stimulation generated by electroporation leads to an increase in cell membrane permeability and the introduction of transfectants into cells [7]. However, electrical stimulation can induce cell injury or toxicity via cell membrane damage, potentially affecting gene expression. Therefore, optimization of electroporation conditions is critical for balancing transfection efficiency and cell viability [7,8,9]. For genome editing, the conditions for electroporation must be evaluated before determining the editing efficiency of the target gene.

Platelet-derived growth factor receptor alpha (*PDGFRA*) is a crucial driver gene in glioblastoma (GBM) [10]. Researchers in precision medicine continually detect new variants of uncertain significance (VUS) in *PDGFRA* and other potentially pathogenic genes using next-generation sequencing (NGS) [10,11]. One approach for analyzing VUS involves functional analysis of a cancer cell line harboring variants generated by genome editing, which has proven useful in the study of *PDGFRA* [12,13]. In our previous genome editing studies using electroporation, we observed that the electroporation process unexpectedly decreased the expression of *PDGFRA* mRNA in GBM cell lines [12]. We speculated that the decrease was due to physical cell membrane damage by electroporation; however, the reason was not clear. Accordingly, we attempted to evaluate the relationship between the decrease in *PDGFRA* mRNA expression and electroporation in this study. This adverse effect can lead to overestimation or underestimation of knockout gene functions or even erroneously indicate the success of gene knockout. Therefore, it is essential to understand how electroporation influences gene expression, particularly that of key genes such as *PDGFRA*, and to explore alternative transfection methods that might minimize such adverse outcomes.

In this study, we investigated the background effects of electroporation, lipofection, and recombinant adeno-associated virus (rAAV) transfection methods on the expression of *PDGFRA* and receptor tyrosine kinase (*RTK*) genes in human GBM cell lines U-251 MG and U-87 MG and human chronic myelogenous leukemia HAP1 cells to determine the optimal experimental conditions for genome editing. Additionally, we assessed the effects of electroporation and rAAV transfection on gene expression using comprehensive mRNA expression through RNA sequencing (RNA-seq) analysis. The expression of inflammatory cytokines was also evaluated to monitor inflammatory or immune responses after rAAV transfection. To the best of our knowledge, few reports have demonstrated the occurrence of unfavorable alterations in gene expression after electroporation. Here, we report adverse effects of electroporation on gene expression and the necessity of a sufficient culture period after electroporation for cell damage recovery or an alternative rAAV transfection method to lessen the adverse effects.

## 2. Results

### 2.1. Effects of Electroporation on PDGFRA mRNA and PDGFRα Expression and Cell Proliferation

First, we investigated the effects of electroporation on *PDGFRA* mRNA expression using U-251 MG cells, a human GBM cell line used in previous studies [12]. *PDGFRA* mRNA expression decreased two days after electroporation (Day 2) with and without Cas9 ribonucleoprotein (RNP) complex transfection (Figure 1A). To perform a detailed analysis of this decrease, *PDGFRA* mRNA and protein expression levels were examined in U-251 MG, U-87 MG, and HAP1 cells. U-87 MG cells were selected as the human GBM cell line harboring two copies of *PDGFRA* (Figure 1B). HAP1 cells are a near-haploid cell line derived from human chronic myelogenous leukemia and are frequently used for genome editing [14,15,16]. We confirmed that HAP1 cells harbored almost one copy of *PDGFRA* (Figure 1B). *PDGFRA* mRNA expression in U-251 and U-87 MG cells decreased on Day 2 and recovered after Day 13 (Figure 1C). In contrast, *PDGFRA* mRNA expression levels in HAP1 cells did not change during the 13 days of electroporation (Figure 1C). PDGFRα expression decreased in U-251 and U-87 MG cells on Day 2 without Cas9 RNP complex transfection (Figure 1D,E). This decrease lasted up to Day 13 in the U-251 MG cells; however, PDGFRα expression recovered in the U-87 MG cells. PDGFRα expression levels in the HAP1 cells did not change during the 13 days of electroporation (Figure 1D,E). An additional 1-week culture was conducted in U-251 MG cells because these cells did not show full recovery of PDGFRα expression even after Day 13. On Day 21, *PDGFRA* mRNA and protein expression levels were comparable between the control and electroporation groups (Figure 1F–H). Furthermore, cell proliferation was suppressed on Day 2; however, no differences were observed between weeks 2 and 3 (Figure 1I).

### 2.2. RNA Sequencing Analysis After Electroporation in U-251 MG Cells

To further evaluate the effects of electroporation on GBM cells, we performed RNA-seq on U-251 MG cells, with or without electroporation, in the absence of transfectants on Day 2. Using principal component analysis, RNA-seq data obtained from electroporated U-251 MG cells were clustered separately from those obtained without electroporation (Figure 2A). K-means clustering of the RNA-seq data revealed that the mRNA expression profile changed after electroporation (Figure 2B), indicating that electroporation modified the mRNA expression profile of U-251 MG cells. In the differentially expressed gene (DEG) analysis, 561 upregulated and 317 downregulated genes were identified (Figure 2C,D).

The Gene Ontology biological process terms that were enriched in the expressed genes are listed in Table 1. Among the downregulated genes, those associated with pathways related to the cell membrane and cell surface were significantly enriched, including *PDGFRA*. According to differential expression analysis (Table 2), *PDGFRA* mRNA expression decreased after electroporation, as reflected in the results of quantitative reverse transcription polymerase chain reaction (RT-qPCR) analysis (Figure 1C). Among other *RTKs*, mRNA expression of *ERBB2* decreased, whereas that of *KDR* and *MET* increased after electroporation. These changes were confirmed using RT-qPCR and Western blot analyses (VEGFR2 protein is the translation product of *KDR*) on Day 2 (the same day RNA-seq was conducted) and recovered on Day 13 after electroporation (Figure 2E–G). These results indicated that the effects of electroporation on mRNA expression were not limited to *PDGFRA* but were common among various *RTK* genes.

### 2.3. Effects of Transfection Methods on PDGFRA mRNA Expression and Modification Rate in U-251 MG Cells

Electrical stimulation via electroporation induces cell membrane damage and subsequent cell toxicity [7,8,9]. Thus, we speculated that optimizing the electroporation conditions could improve *PDGFRA* mRNA expression. Moderate electroporation conditions, characterized by a lower voltage, led to the recovery of *PDGFRA* mRNA expression in U-251 MG cells on Day 2. However, the editing activity shown by the modification rate (number of reads with modifications [insertion, deletion, and substitution]/number of total reads measured by NGS) introduced by the Cas9 RNP complex markedly decreased under moderate electroporation conditions (Figure 3A,B).

In addition, we investigated the effects of lipofection- and virus-mediated transfection of the Cas9 RNP complex on *PDGFRA* mRNA expression. *PDGFRA* mRNA expression decreased on Day 2 after lipofection with or without the Cas9 RNP complex (Figure 3C), and editing activity (modification rate) was detected only after Cas9 RNP complex transfection (Figure 3D). Downregulation of *PDGFRA* mRNA expression recovered on Day 13 (Figure 3E). The decrease in PDGFRα expression persisted until Day 13 after lipofection in U-251 MG cells; however, this reduction was not observed on Day 21 (Figure 3F,G). Changes in *PDGFRA* mRNA and protein expression levels after lipofection were similar to those observed after electroporation. Furthermore, we measured *ERBB2*, *KDR*, and *MET* mRNA expression levels, which were altered after electroporation. *ERBB2* mRNA expression decreased, and that of *KDR* and *MET* increased on Day 2, which were all restored to control levels on Day 13 post-electroporation (Figure 3H). These changes in mRNA expression of *RTKs* after lipofection were comparable to those observed after electroporation.

### 2.4. Effects of Recombinant Adeno-Associated Virus Transduction on PDGFRA mRNA Expression and Modification Rate in U-251 MG Cells

Because virus-mediated Cas9 transfection is available for genome editing [18,19], we used a Cas9 rAAV for Cas9 delivery. The packaging capacity of the rAAV is approximately 4.5 kb [19], and a small Cas9 ortholog [*Staphylococcus aureus* (SaCas9), approximately 3.1 kb] was used for rAAV construction in this study. *PDGFRA* mRNA expression levels were not affected on Day 2 after rAAV transfection (Figure 4A), and the modification rate increased in a dose-dependent manner (MOI) (Figure 4B). No change in the mRNA expression of *RTKs* was observed after rAAV infection (Figure 4C). To confirm that the SaCas9 protein did not affect mRNA expression of *PDGFRA* and *RTKs*, mRNA expression levels were measured on Day 2 after mNeonGreen fluorescent protein-expressing rAAV infection at the same MOI, which revealed no changes in mRNA expression (Figure 4D,E).

### 2.5. RNA Sequencing Analysis After rAAV Infection in U-251 MG Cells

Viral transfection is a highly efficient gene transfer method; however, it can result in immunogenicity [20]. Because viral transfection is a different transfection method than electroporation, which is a physical transfection method, we speculated that the influence of viral transfection on cells may be distinct from that of electroporation and lipofection. To address this, we conducted RNA-seq analysis on Day 2 after rAAV infection in U-251 MG cells (at the same time point as RNA-seq after electroporation). Although RNA-seq data were clustered separately between groups with and without rAAV infection using principal component analysis (Figure 5A), k-means clustering did not cluster the RNA expression profiles with or without rAAV transfection (Figure 5B). Only 42 upregulated and 25 downregulated genes were identified in DEG analysis (Figure 5C,D). The number of differentially expressed genes after rAAV infection was lower than that after electroporation. In the enrichment analysis, only four significant pathways were detected in the downregulated pathway, whereas no significant enrichment was detected in the upregulated pathway. The altered gene numbers related to pathways in the cells with rAAV transfection (Table 3) were fewer than those in the cells with electroporation (Table 1), indicating that rAAV transfection had less effect on the RNA-seq profile. We analyzed the RNA-seq results in detail and found that *CCL2*, which is related to the immune response, was among the downregulated genes. To further elucidate the effects of rAAV infection on the immune response, *CCL2* mRNA expression in U-251 MG cells was measured using RT-qPCR. Subsequently, we measured the expression levels of the major inflammatory cytokines (*IL1B, IL6,* and *CXCL8*) [21] (Figure 5E). Elevated *IL1B* mRNA expression was detected on Day 1 after rAAV infection; however, no significant difference was observed on Day 2. Although downregulation of *IL6* and *CCL2* mRNA expression was detected on Day 2 in both RNA-seq and RT-qPCR analyses, no changes were observed at other time points. No effect on *CXCL8* mRNA expression was observed in U-251 MG cells until Day 13.

## 3. Discussion

The present study showed that electroporation itself carries the risk of unintended alterations in gene expression; therefore, it is critical to determine the optimal conditions for Cas9 transfer to ensure successful gene editing. Cas9-based genome editing technologies are increasingly being used in various fields, including basic research, with electroporation being commonly employed for Cas9 expression plasmids and protein transfection. Electroporation has also been widely utilized in other areas of molecular biology research [6,7], highlighting the necessity of elucidating its effects on gene expression and determining the optimal electroporation conditions.

In this study, we investigated the effects of electroporation on *PDGFRA* mRNA and protein expression in cultured human GBM and HAP1 cells. We found that *PDGFRA* mRNA and protein expression levels decreased unexpectedly after electroporation without the transfection of the Cas9 RNP complex. This effect likely stems from cell membrane damage induced by electroporation [7,8,9], as alterations in the membrane structure may influence the expression of genes such as *PDGFRA*. Gene expression is known to change when cell membranes are altered by the delivery of nanoparticles [22], and similar structural or functional changes may occur after electroporation. RNA-seq analysis further revealed that the mRNA expression levels of other *RTKs* also varied after electroporation, suggesting that electroporation-induced changes in mRNA expression are not uniform and can unpredictably affect gene expression. This underscores the importance of verifying the effects of electroporation on target genes before proceeding with genome editing, as such as alterations may undermine the intended gene knockout effects or lead to misinterpretation of successful gene editing. On the other hand, *PDGFRA* mRNA expression levels in HAP1 cells did not change after electroporation. Although this might be characteristic of HAP1 cells (e.g., ploidy of the cell and resistance for the electroporation), the reason for no change in *PDGFRA* mRNA expression in HAP1 cells is not clear in this study.

We also observed that while PDGFRα expression was restored 13 days after electroporation in U-87 MG cells, a longer recovery period (approximately one additional week) was required for PDGFRα expression to return to baseline in U-251 MG cells. This difference in recovery time could be related to recovery from cell damage. To explore this further, we conducted a cell proliferation assay in U-251 MG cells and found that the growth rate of these cells was similar to that of control cells, except at 2 days post-electroporation. These results suggest that electroporation-induced cell injury likely involves complex biochemical mechanisms [8] that reduce PDGFRα expression, independent of cell proliferation. Therefore, cell recovery should not be solely assessed via proliferation assays after electroporation but should instead be evaluated in conjunction with protein expression analyses to obtain a more accurate assessment of recovery. In general, the necessity for a sufficient recovery period after electroporation might not be critical, because most gene editing studies require several weeks for cell culture, damage recovery, and cloning. In contrast, this is particularly important in saturation genome editing studies, which often analyze gene-edited pools of cells without cloning. To minimize the risk of erroneous conclusions, we performed control experiments using cells treated solely with electroporation. This allowed us to compare mRNA and protein expression levels at each experimental time point between the treated and control cells. However, it is important to note that there is limited documentation of this fundamental principle in electroporation-based experiments, and this gap in the literature should be addressed in future studies.

In U-251 MG cells, the effects on *PDGFRA* and *RTK* mRNA expression after lipofection were similar to those observed after electroporation. However, rAAV transfection did not affect *PDGFRA* and *RTK* mRNA expression and had less of an impact on the RNA-seq profile. Additionally, rAAV transfection only slightly affected the expression of inflammation- or immune-related genes, in contrast with the known inflammatory and immune responses induced by biological transfection using other recombinant viral vectors [21]. These findings are in line with a previous report suggesting that immune system activation is absent in *in vitro* but moderate in *in vivo* studies using rAAV vectors compared to the more pronounced immune responses observed in both *in vitro* and *in vivo* studies with recombinant adenovirus vectors [23]. Overall, rAAV-mediated transfection was less disruptive to baseline gene expression and caused fewer inflammatory and immune responses. This may be due to the less harmful biological gene delivery mechanism of viral vector transfer [24], which differs from the more disruptive physical and chemical methods of electroporation and lipofection [20]. Based on these findings, we concluded that rAAV transfection is the most suitable method for genome editing among the conditions tested.

A potential limitation of the present study is that it primarily focused on *PDGFRA* expression in GBM and HAP1 cells. Although the results indicated that similar unexpected effects of electroporation may occur in other genes and cells of interest, the scope of this study was limited to these specific contexts. Moreover, in other cell types, the optimal conditions for successful genome editing remain uncertain and controversial [13], suggesting that the appropriate selection of transfection methods and cells is essential for successful genome editing. Furthermore, although we identified some discrepancies between mRNA and protein expression in U-251 MG cells, these were not observed in U-87 MG or HAP1 cells after electroporation. Such discrepancies between mRNA and protein expression can occur due to the post-translational regulation of protein expression [25,26], but the mechanisms underlying these differences were not fully elucidated in this study. Future studies should explore these mechanisms further. In general, HAP1 or U-87 MG cells appear to be better suited for studies utilizing electroporation-mediated gene transfer, whereas rAAV transfection may be more appropriate for experiments that require minimal disruption of cellular function.

In conclusion, the effects of electroporation on gene expression are not uniform and vary depending on cell type and target gene. Therefore, we recommend that the choice of transfection method and the potential alterations in target gene expression be thoroughly confirmed before initiating genome editing studies using Cas9-based technologies. This approach will help avoid erroneous evaluations of gene knockout success or unintended effects on gene expression, ensuring the reliability of the experimental outcomes. Nevertheless, further research is needed to refine transfection protocols and understand the broader implications of electroporation and other gene delivery methods for genome editing.

## 4. Materials and Methods

### 4.1. Cell Culture

U-251 MG cells were obtained from the JCRB Cell Bank (Osaka, Japan) and U-87 MG cells were purchased from the American Type Culture Collection (Manassas, VA, USA). HAP1 cells were purchased from Horizon Discovery Ltd. (Cambridge, UK). HEK293T cells were purchased from Takara Bio Inc. (Shiga, Japan). U-251 and U-87 MG cells were maintained in Eagle’s minimum essential medium (FUJIFILM Wako Pure Chemical, Osaka, Japan), HAP1 cells were maintained in Iscove’s Modified Dulbecco’s medium (FUJIFILM Wako), and HEK293T cells were maintained in Dulbecco’s Modified Eagle Medium (FUJIFILM Wako) at 37 °C with 95% air and 5% CO_2_. The culture medium was supplemented with 2 mM glutamine, 100 U/mL penicillin, 100 μg/mL streptomycin, and 10% fetal bovine serum (Sigma-Aldrich, St Louis, MO, USA).

### 4.2. Fluorescence In Situ Hybridization

Sections (4 μm thick) were cut from formalin-fixed, paraffin-embedded cell blocks for fluorescence in situ hybridization analysis. The sections were immersed in 0.2 N HCl for 20 min, then in distilled water for 3 min, and finally in 2× saline–sodium citrate buffer (SSC). Slides were microwaved in a pressure jar and digested with protease I (Abbott Laboratories, Des Plaines, IL, USA) for 45 min. The slides were washed twice with 2× SSC, air-dried, fixed with phosphate-buffered neutral 10% formalin for 10 min, and dehydrated in ethanol. Artificial bacterial chromosome clones RP11-231C18 (red) and CEP4 (green) probes (Empire Genomics, Buffalo, NY, USA) were used to analyze the *PDGFRA* ploidy. The sections were then transferred to ThermoBrite (Abbott), which was programmed to perform denaturation at 85 °C for 1 min, followed by hybridization at 37 °C for 16 h. The slides were then air-dried in the dark and counterstained with 4,6-diamidino-phenyl-indole. Images were captured using a fluorescence microscope (BX51; Olympus, Tokyo, Japan), and the red and green fluorescence signals in each section were counted in 30 cells.

### 4.3. Electroporation

Cells were trypsinized and suspended in Opti-MEM I reduced-serum medium (Thermo Fisher Scientific, Waltham, MA, USA) at a concentration of 1–2 × 10^7^ cells/mL. The Cas9 RNP complexes were added to 80 μL of the cell suspension, and the mixture was transferred to 2-mm cuvettes (Nepa Gene, Chiba, Japan). For RNP complex electroporation, an electroporation enhancer (1.2 μM, Integrated DNA Technologies (IDT), Coralville, IA, USA) was added to the cell suspension. Electroporation was conducted using a NEPA21 electroporator (Nepa Gene) with 2× poring pulses (GBM cells: voltage 150 V; length 7.5 ms; interval 50 ms; polarity +, HAP1 cells: voltage 275 V; length 0.5 ms; interval 50 ms; polarity +) and 5× transfer pulses (voltage: 20 V; length: 50 ms; interval: 50 ms; polarity ±). Electroporated cells were seeded in 6-well plates (Thermo Fisher Scientific). Cas9 RNP complex formation was performed using the following procedure: equimolar amounts of CRISPR RNA (crRNA) and trans-activating crRNA (tracrRNA, IDT) were hybridized for 5 min at 95 °C to form a single-guide RNA. Subsequently, the single-guide RNA (1.2 μM) and Cas9 nuclease (1 μM; IDT) were mixed with Opti-MEM to form ribonucleoprotein complexes, which were then incubated for 30 min at room temperature. The crRNA sequence has been described previously [12,13].

### 4.4. Quantitative Reverse Transcription–Polymerase Chain Reaction

Cells were harvested on the indicated days, and their total RNA was extracted using an RNeasy kit (QIAGEN, Hilden, Germany) and converted into complementary DNA using ReverTra Ace qPCR RT Master Mix (TOYOBO, Osaka, Japan). The complementary DNA (5–10 ng/reaction) was mixed with 2× master mix (THUNDERBIRD Probe qPCR Mix, TOYOBO), probe (4 pmol), and a primer pair (6 pmol each) according to the manufacturer’s instructions and amplified using LightCycler 480 (Roche Diagnostics, Basel, Switzerland) under the following cycling conditions: 1 cycle at 95 °C for 60 s, then 40 cycles at 95 °C for 15 s and 60 °C for 60 s. The premixed probe and primer pair (PrimeTime qPCR Probe Assays, IDT) were used as follows: *PDGFRA*: Hs.PT.58.45699973; *ERBB2*: Hs.PT.58.1330269; *KDR*: Hs.PT.58.3285240; *MET*: Hs.PT.58.339430; *IL1B*: Hs.PT.58.1518186; *IL6*: Hs.PT.58.40226675; *CXCL8*: Hs.PT.58.39926886.g; *CCL2*: Hs.PT.58.45467977. Each sample was analyzed in triplicate in separate wells for each target. The average of the three threshold cycle values was calculated for the target and reference genes and analyzed using the comparative Ct method. The Ct values were normalized to a reference gene (18S rRNA, probe and primer pair: Hs.PT.58.14390640).

### 4.5. Western Blotting

Cells were washed with phosphate-buffered saline (PBS) and precipitated with 10% trichloroacetic acid on ice for 30 min. The precipitate was washed with cold PBS and dissolved in cold lysis buffer (7 M urea, 2 M thiourea, 3% CHAPS, and 1% Triton X-100). The lysates were fractionated using sodium dodecyl sulfate–polyacrylamide gel electrophoresis and transferred onto polyvinylidene difluoride membranes. The membranes were blocked with 5% nonfat dry milk in Tris-buffered saline containing 0.1% Tween 20 and incubated overnight at 4 °C with the relevant primary antibodies diluted in Can Get Signal Solution 1 (TOYOBO). Subsequently, the membranes were incubated for 1 h at room temperature with HRP-conjugated anti-rabbit IgG antibody (#7074; Cell Signaling Technology, Danvers, MA, USA) or HRP-conjugated anti-mouse IgG antibody (#7076; Cell Signaling Technology), and proteins were detected using Clarity Max Western ECL Substrate (Bio-Rad Laboratories, Hercules, CA, USA) or SuperSignal West Pico chemiluminescent substrate (Thermo Fisher Scientific). The proteins were detected using a chemiluminescence imaging system (Ez-Capture MG, ATTO, Tokyo, Japan). The following primary antibodies were used: rabbit anti-PDGFRα monoclonal (#3174; Cell Signaling Technology), rabbit anti-erbB-2 monoclonal (#2165; Cell Signaling Technology), rabbit anti-VEGF receptor 2 monoclonal (#9698; Cell Signaling Technology), rabbit anti-MET monoclonal (#8198; Cell Signaling Technology), and mouse anti-β-actin (sc-47778; Santa Cruz Biotechnology, Dallas, TX, USA).

### 4.6. Cell Proliferation Assay

Cells were seeded at 2.5 × 10^3^ cells/well in 96-well plates (Thermo Fisher Scientific, Waltham, MA, USA) with medium supplemented with 10% FBS and allowed to attach overnight. The Cell Counting Kit-8 (WST-8; Dojindo, Kumamoto, Japan) was used to count viable cells according to the manufacturer’s instructions on the indicated days. Cell viability was determined by measuring the amount of formazan dye generated at 450 nm against a reference wavelength at 620 nm using a microplate reader (Thermo Fisher Scientific).

### 4.7. RNA Sequencing

Cells were harvested 2 days after electroporation or rAAV infection, and their total RNA was extracted as described above. Pair-end RNA-seq was performed using NovaSeq 6000 (Illumina, San Diego, CA, USA) on Rhelixa (Tokyo, Japan). RNA-seq analysis was performed as follows: raw sequence reads were trimmed and filtered using PrinSeq-lite (version 0.20.4) [27]. The trimmed reads were aligned using HISAT2 (version 2.2.1) [28]. FeatureCounts (version 2.0.6) [29] was used to quantify the expression of each gene. The iDEP application [17] was used to analyze and visualize changes in gene expression.

### 4.8. Lipofection

A mixture of Cas9 RNP complex (1 μM) and 1.2 μL of Lipofectamine RNAiMAX reagent (Thermo Fisher Scientific) was transferred to the wells of 96-well plates (Thermo Fisher Scientific), then 100 μL of cell suspensions at 1 × 10^5^ cells/mL were added to the wells containing the transfection complex. After the 2-day culture, the cells were harvested, and their genomic DNA was extracted using a Wizard SV Genomic DNA Purification System (Promega, Madison, WI, USA). The total RNA was extracted as described above.

### 4.9. Viral Production, Purification, Titering, and Infection

CRISPR/SaCas9 virus was prepared using a triple-transfection, helper-free method with serotype 2 packaging. The AAV vector (pX601-AAV-CMV::NLS-SaCas9-NLS-3xHA-bGHpA; U6::BsaI-sgRNA, #61591, Addgene, Watertown, MA, USA) and pRC and pHelper vectors (CRISPR/SaCas9 Helper Free System (AAV2), Takara Bio) were transfected into HEK293T cells using the XFect Transfection Reagent (Takara Bio) according to the manufacturer’s instructions for rAAV production. The cells were cultured with medium containing 2% FBS and harvested 3 days post-transfection. The rAAV was purified using an AAV purification kit (Takara Bio). The purified rAAV was titered using the quantitative PCR-based method reported by Ahammer [30]. The sequence of the single-guided RNA was “acggagatccactcccgagac”, and oligonucleotides for the single-guided RNA were annealed and cloned into the AAV vector, as reported by Ran [19]. The mNeonGreen fluorescent protein-expressing rAAV was produced from pAAV-CAG-mNeonGreen (#99134, Addgene). The cells were seeded at a density of 2 × 10^4^ cells/well in 48-well plates (Greiner, Kremsmünster, Austria) and infected with rAAV at the indicated multiplicity of infection on the following day. After two days of culturing, the cells were harvested, and their genomic DNA and total RNA were extracted as described above.

### 4.10. High-Throughput Sequencing

The genomic DNA was sequenced using a MiSeq sequencer (Illumina). The genomic region of interest was amplified using primers containing Illumina forward and reverse adapters (forward primer: 5′- TCTTTCCCTACACGACGCTCTTCCGATCTGCAAACCTTAGAGGTTCTGGCAAGGAG, reverse primer: 5′-GTGACTGGAGTTCAGACGTGTGCTCTTCCGATCTACCTTATATTCCCAGGGCCGGTTAATG) via PCR. PCR was performed with genomic DNA (50–150 ng) using the KOD One PCR Master Mix (TOYOBO) and conducted as follows: 30 cycles at 98 °C for 10 s, 67 °C for 5 s, and 68 °C for 10 s. Unique Illumina barcoding primer pairs were added to each sample during the second PCR step. The second PCR step was conducted with 1 μL of the unpurified first PCR reaction mixture using the KOD One PCR Master Mix as follows: 10 cycles of 98 °C for 10 s and 68 °C for 10 s. The PCR products were monitored using a Microchip Electrophoresis System (MultiNA, Shimadzu Corporation, Kyoto, Japan). DNA libraries were prepared using mixtures of equal molar amounts of PCR products and purified using solid-phase paramagnetic beads (AMPure XP; Beckman Coulter, Brea, CA, USA). DNA concentration was measured using fluorometric quantification (Qubit; Thermo Fisher Scientific). The alignment of amplicon sequences to a reference sequence and quantification of editing efficiency were performed using CRISPResso2 software (version 2.3.2) [31].

### 4.11. Statistical Analysis

Data were plotted and analyzed using R (version 4.1.0) and ggplot2 (version 3.3.3). Three or more replicates were performed for all experiments, and all data are presented as means ± standard error. Statistical significance was determined using a one-way analysis of variance (ANOVA) with Tukey’s post hoc test, and the results were considered statistically significant at *p* < 0.05.

## Figures and Tables

**Figure 1 cimb-47-00091-f001:**
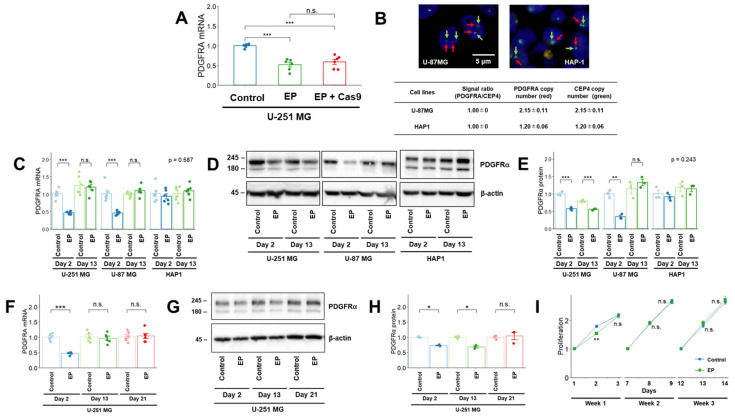
Effects of electroporation on *PDGFRA* mRNA and PDGFRα expression. (**A**) Effect of electroporation on *PDGFRA* mRNA expression in U-251 MG cells. (**B**) Upper panels: Representative images of the fluorescence in situ hybridization analysis of U-87 MG and HAP1 cells. A probe mix for *PDGFRA* (red arrows, BAC clone RP11-231C18) and CEP4 (green arrows) was used. Scale bar: 5 μm. Lower panel: *PDGFRA* and CEP4 signals were calculated by counting the number of signals from 30 cells. The CEP4 signal was used to determine chromosome copy number (ploidy status). The *PDGFRA*/CEP4 signal ratio was calculated using the following formula: signal ratio = (*PDGFRA* signal)/(CEP4 signal). (**C**,**D**) Effects of electroporation on *PDGFRA* mRNA (**C**) and PDGFRα (**D**) expression in U-251 MG, U-87 MG, and HAP1 cells. (**E**) Densitometric quantification of the Western blot results in Figure 1D. (**F**,**G**) Effects of electroporation on *PDGFRA* mRNA (**F**) and PDGFRα (**G**) expression in U-251 MG cells. (**H**) Densitometric quantification of the Western blot results in Figure 1G. (**I**) Cell proliferation was monitored after electroporation in U-251 MG cells using WST-8 assay. Values are presented as fold-increases over those on the day following seeding (n = 6). The day of electroporation was designated as Day 0. Quantitative data are represented as mean ± standard error. *, *p* < 0.05; **, *p* < 0.01; ***, *p* < 0.001; n.s., not significant. *PDGFRA*, platelet-derived growth factor receptor alpha; EP, electroporation; CEP4, chromosome enumeration probe for chromosome 4.

**Figure 2 cimb-47-00091-f002:**
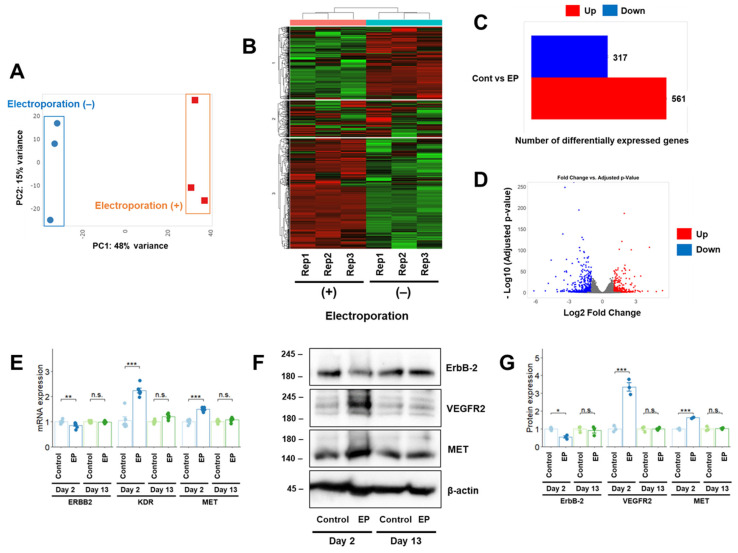
RNA-seq analysis of U-251 MG cells after electroporation. (**A**–**D**) Principal component analysis (**A**), k-means clustering analysis (**B**), and DEG analyses, (**C**,**D**) transcript counts derived from U-251 MG cells treated with or without electroporation using the iDEP application [17]. (**E**,**F**) Effects of electroporation on *RTK* mRNA (**E**) and protein (**F**) expression in U-251 MG cells. (**G**) Densitometric quantification of the Western blot results in (**F**). The day of electroporation was designated as Day 0. Quantitative data represented as mean ± standard error. *, *p* < 0.05; **, *p* < 0.01; ***, *p* < 0.001; n.s., not significant. Rep1, Rep2, and Rep3 represent the three replicates. EP, electroporation.

**Figure 3 cimb-47-00091-f003:**
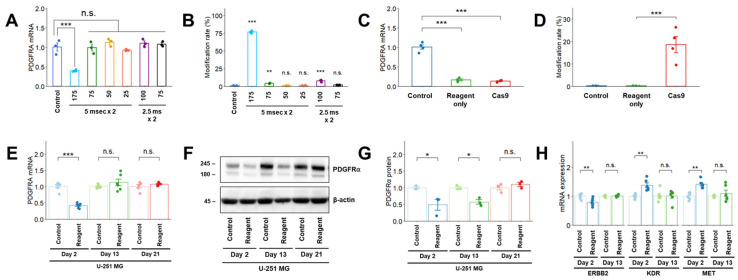
(**A**–**D**) Variation in *PDGFRA* mRNA expression after electroporation under various conditions (**A**) and lipofection (**C**) on Day 2 after transfection in U-251 MG cells. Editing activity (modification rate) after electroporation using modified conditions (**B**) and lipofection (**D**) on Day 2 after transfection in U-251 MG cells. The modification rate was calculated as the number of reads with modifications (insertion, deletion, and substitution) divided by the number of total reads measured via NGS. (**E**,**F**) Effects of lipofection on *PDGFRA* mRNA (**E**) and PDGFRα (**F**) expression in U-251 MG cells. (**G**) Densitometric quantification of the Western blot results in (**F**); (**H**) Effects of lipofection on *RTK* mRNA expression 2 days after transfection in U-251 MG cells. The day of transfer (electroporation and lipofection) was designated as Day 0. Quantitative data are presented as mean ± standard error. *, *p* < 0.05; **, *p* < 0.01; ***, *p* < 0.001; n.s., not significant.

**Figure 4 cimb-47-00091-f004:**
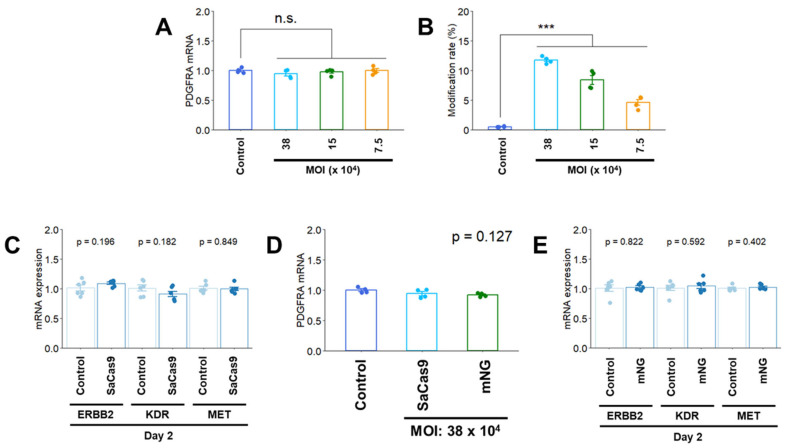
(**A**) *PDGFRA* mRNA expression on Day 2 after rAAV transfection in U-251 MG cells. (**B**) Editing activity (modification rate) on Day 2 in U-251 MG cells. The modification rate is calculated as the number of reads with modifications (insertion, deletion, and substitution)/number of total reads measured by NGS. (**C**) Effects of rAAV infection on *RTK* mRNA expression in U-251 MG cells. (**D**,**E**) Effects of infection with mNeonGreen (mNG) fluorescent protein-expressing rAAV on *PDGFRA* and *RTK* mRNA expression in U-251 MG cells. The day of rAAV infection was designated as Day 0. Quantitative data are represented as mean ± standard error. ***, *p* < 0.001; n.s., not significant.

**Figure 5 cimb-47-00091-f005:**
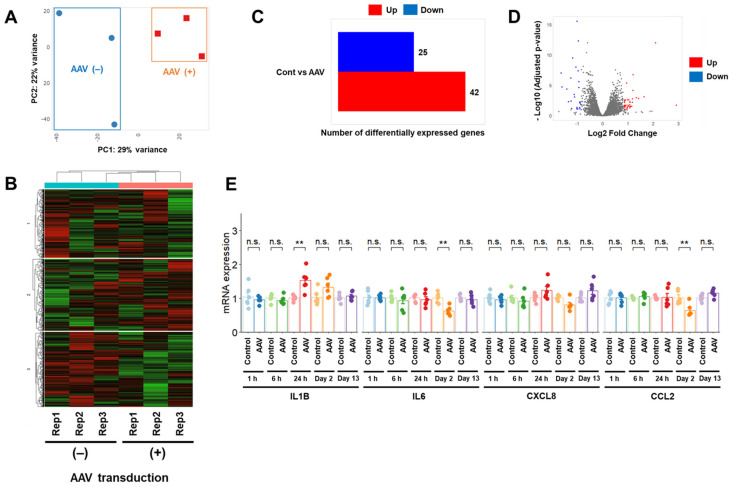
RNA-seq analysis after rAAV transduction in U-251 MG cells. (**A**–**D**) Principal component (**A**), k-means clustering (**B**), and DEG analyses (**C**,**D**) for transcript counts derived from U-251 MG cells treated with or without rAAV infection using iDEP [17]. (**E**) Effects of rAAV infection on mRNA expression of immune-related genes in U-251 MG cells. The day of rAAV infection was designated as Day 0. Quantitative data are presented as mean ± standard error. **, *p* < 0.01; n.s., not significant. Rep1, Rep2, and Rep3 represent the three replicates.

**Table 1 cimb-47-00091-t001:** Enriched Gene Ontology cellular component terms in up- and downregulated genes (electroporation).

Direction	Pathways	Number of Genes	Adjusted *p*-Value
Down	Integral component of plasma membrane *	134	6.93 × 10^−12^
	External encapsulating structure	70	8.02 × 10^−12^
	Intrinsic component of plasma membrane *	138	8.02 × 10^−12^
	Collagen-containing extracellular matrix	54	1.40 × 10^−8^
	Plasma membrane region	107	1.52 × 10^−5^
	Receptor complex *	43	4.22 × 10^−5^
	Synapse	116	7.97 × 10^−5^
	Cell junction *	170	9.86 × 10^−5^
	Cell projection *	184	0.000102
	Plasma membrane-bounded cell projection *	177	0.00011
	Basolateral plasma membrane	30	0.000115
	Cell surface *	65	0.00014
	Basal plasma membrane	32	0.000156
	Extracellular region	253	0.000158
Up	Intrinsic component of plasma membrane	84	3.20 × 10^−6^
	Integral component of plasma membrane	77	2.44 × 10^−5^
	Replication fork	13	0.000863
	Ctf18 RFC-like complex	5	0.000883
	Nuclear replication fork	8	0.006975

* Pathways containing *PDGFRA* genes.

**Table 2 cimb-47-00091-t002:** Differential expression analysis of receptor tyrosine kinases (electroporation).

Genes	Log Fold Changes	*p* Value
*PDGFRA*	–1.07322	1.04 × 10^−47^
*ERBB2*	–1.10129	1.54 × 10^−19^
*KDR*	2.077597	1.09 × 10^−53^
*MET*	1.048319	3.57 × 10^−52^

**Table 3 cimb-47-00091-t003:** Enriched Gene Ontology cellular component terms in up- and down-regulated genes (rAAV infection).

Direction	Pathways	Number of Genes	Adjust *p*-Value
Down	External encapsulating structure	5	1.95 × 10^−2^
	Extracellular matrix	5	1.95 × 10^−2^
	Extracellular region	11	5.18 × 10^−2^
	Collagen-containing extracellular matrix	4	5.18 × 10^−2^
Up	No significant enrichment found		

## Data Availability

The data presented in this study are available from the corresponding author upon request.

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
