# Peer review of "Electroporation Induces Unexpected Alterations in Gene Expression: A Tip for Selection of Optimal Transfection Method"

_cimb, 2025, doi:10.3390/cimb47020091_

Round 1
Reviewer 1 Report
Comments and Suggestions for Authors
This manuscript presents data on the effects of electroporation (and other gene transfer techniques) on the expression of selected genes. The results show that electroporation significantly influences the expression levels of some genes; this is obviously important to know in interpreting experiments using electroporation to introduce gene-editing reagents (and other reagents) into cells. In general the results are straightforward and clearly described. However, I do not fully understand some key experiments. Fig. 1A and several other figures compare mRNA levels for a specific gene under several conditions. I assume that the mRNA levels were quantified by RT-PCR as in lines 359-369. But how were they normalized? Were the analyses performed on equal amounts of total RNA? If so, what was this amount? Or are they normalized against the copy numbers of 18s rRNA, which apparently was also measured?
A minor point: I imagine the authors meant “nucleic acid” rather than “nucleotides” in line 42.
Author Response
|
Response to Reviewer 1 Comments
|
||
|
1. Summary |
|
|
|
We would like to thank Reviewer 1 for the helpful and constructive comments on our study. We revised the manuscript following the detailed feedback provided by the Reviewer. We hope that the Reviewer will find our responses below properly address all the raised concerns. |
||
|
2. Questions for General Evaluation |
Reviewer’s Evaluation |
Response and Revisions |
|
Does the introduction provide sufficient background and include all relevant references? |
Yes |
|
|
Is the research design appropriate? |
Yes |
|
|
Are the methods adequately described? |
Can be improved |
We revised the material and methods section. |
|
Are the results clearly presented? |
Yes |
|
|
Are the conclusions supported by the results? |
Yes |
|
|
3. Point-by-point response to Comments and Suggestions for Authors |
||
|
Comments 1: Fig. 1A and several other figures compare mRNA levels for a specific gene under several conditions. I assume that the mRNA levels were quantified by RT-PCR as in lines 359-369. But how were they normalized? Were the analyses performed on equal amounts of total RNA? If so, what was this amount? Or are they normalized against the copy numbers of 18s rRNA, which apparently was also measured? |
||
|
Response 1: We agree with this comment. The section on RT-qPCR methods was poorly described, and we additionally mentioned it in detail (lines 371 to 377, 382 to 383). |
||
|
Comments 2: I imagine the authors meant “nucleic acid” rather than “nucleotides” in line 42. |
||
|
Response 2: We agree with the Reviewer and incorporated this suggestion in the revised manuscript (line 42). |
||
Reviewer 2 Report
Comments and Suggestions for Authors
This manuscript discusses that electroporation is a useful technique for delivering nucleotides and proteins, which affects the expression of PDGFRA and receptor tyrosine kinases (RTKs) in U-251 and U-87 MG cells. In U-87 MG cells, the expression of PDGFRA returned to normal after 13 days following electroporation, while U-251 MG cells showed ongoing suppression even though their mRNA levels recovered. These results highlight the importance of allowing enough recovery time or considering other methods, like rAAV transfection, to properly assess the results of CRISPR gene editing. The manuscript is well-organized and includes data from significant and interesting data; however, I recommend that the authors carefully address the following points.
· Major issues
1. The scientific rationale for this study is not clearly articulated. It would be helpful to explicitly and comprehensively outline the motivations for conducting this research in the introduction section. Consider incorporating more references to previous related studies or experimental data as necessary.
2. In lines 55-58, the authors state that the expression of PDGFRA mRNA decreased in a prior study. Is there a published paper or data to support this? It is crucial to clarify why the PDGFRA gene was chosen as the target for investigating the side effects of transfection methods.
3. The HAP1 cell line, which is near haploid, did not exhibit changes in PDGFRA mRNA expression levels. Could this be attributed to its haploid nature? The explanation for the results related to HAP1 is unclear. Please provide a detailed rationale.
4. Although the expression of PDGFRA and RTKs mRNA was not influenced by rAAV infection, Figure 5 indicates that the expression of many genes was altered due to rAAV infection. Can we definitively conclude that rAAV infection did not impact gene expression changes? Please explain the possible reasons for this observation.
· Minor issues
1. In figure 1, it is necessary to mention in the legend that “EP” stands for Electroporation.
2. In figure 2G, correct “erbB-2" to “ErbB-2”
3. On line 275 and 277, “RTKs” to “RTKs”
Author Response
|
Response to Reviewer 2 Comments
|
||
|
1. Summary |
|
|
|
We would like to thank Reviewer 2 for the helpful and constructive comments on our study. We revised the manuscript following the detailed feedback provided by the Reviewer. We hope that the Reviewer will find our responses below properly address all the raised concerns. |
||
|
2. Questions for General Evaluation |
Reviewer’s Evaluation |
Response and Revisions |
|
Does the introduction provide sufficient background and include all relevant references? |
Can be improved |
We revised the introduction section according to the reviewer’s comments. |
|
Is the research design appropriate? |
Yes |
|
|
Are the methods adequately described? |
Yes |
|
|
Are the results clearly presented? |
Yes |
|
|
Are the conclusions supported by the results? |
Can be improved |
We revised the results section. |
|
3. Point-by-point response to Comments and Suggestions for Authors |
||
|
Comments 1 (Major 1): The scientific rationale for this study is not clearly articulated. It would be helpful to explicitly and comprehensively outline the motivations for conducting this research in the introduction section. Consider incorporating more references to previous related studies or experimental data as necessary. |
||
|
Comments 2 (Major 2): In lines 55-58, the authors state that the expression of PDGFRA mRNA decreased in a prior study. Is there a published paper or data to support this? It is crucial to clarify why the PDGFRA gene was chosen as the target for investigating the side effects of transfection methods. |
||
|
Responses 1 and 2: Thank you for pointing this out and we are sorry for lack of explanation. This decrease in PDGFRA mRNA expression was observed and was due to unknown mechanism in our previous study (reference #12). We expected that the decrease was due to physical cell membrane damage caused by electroporation and attempted to identify the relationship between the decrease in PDGFRA mRNA expression and electroporation in this study. We added the description of this point to the manuscript (lines 57 to 60). |
||
|
Comments 3 (Major 3): The HAP1 cell line, which is near haploid, did not exhibit changes in PDGFRA mRNA expression levels. Could this be attributed to its haploid nature? The explanation for the results related to HAP1 is unclear. Please provide a detailed rationale. |
||
|
Response 3: We could not identify the cause of no change in the PDGFRA mRNA expression in HAP1 cells in this study, and could not find article about the relationship between haploid in HAP1 cells and mRNA expression. We added this point to the manuscript in lines 260 to 264. |
||
|
Comments 4 (Major 4): Although the expression of PDGFRA and RTKs mRNA was not influenced by rAAV infection, Figure 5 indicates that the expression of many genes was altered due to rAAV infection. Can we definitively conclude that rAAV infection did not impact gene expression changes? Please explain the possible reasons for this observation. |
||
|
Response 4: Although tens of up- and down-regulated genes were observed after rAAV infection (Figure 5C and 5D); however, the numbers of change in mRNA expression after rAAV infection were apparently lower than that after electroporation, which impacted to the hundreds of mRNA expressions. Furthermore, the number of significant pathways were also a little in enrichment analysis. Commitment of rAAV infection to the significant pathways remains elusive, since only a small number of genes related to the pathways. As these findings, we considered that the effect of rAAV infection on the mRNA expression to cells was slight. We intended to document the above, but the description was unclear. We then added to the description to our manuscript (lines 217 to 219). |
||
|
Comments 5: (Minor 1): In figure 1, it is necessary to mention in the legend that “EP” stands for Electroporation. (Minor 2): In figure 2G, correct “erbB-2" to “ErbB-2”. (Minor 3): On line 275 and 277, “RTKs” to “RTKs” |
||
|
Response 5: We agree with the Reviewer and incorporated this suggestion in the revised manuscript (figures 1 and 2 legend, figure 2, lines 285 and 287). The other “RTKs” in this manuscript also corrected to “RTKs”. |
||